# Enhanced Adhesion of Electrospun Polycaprolactone Nanofibers to Plasma-Modified Polypropylene Fabric

**DOI:** 10.3390/polym15071686

**Published:** 2023-03-28

**Authors:** Lucie Janů, Eva Dvořáková, Kateřina Polášková, Martina Buchtelová, Petr Ryšánek, Zdeněk Chlup, Tomáš Kruml, Oleksandr Galmiz, David Nečas, Lenka Zajíčková

**Affiliations:** 1Plasma Technologies for Materials, Central European Institute of Technology—CEITEC, Brno University of Technology, Purkyňova 123, 612 00 Brno, Czech Republic; 2Department of Condensed Matter Physics, Faculty of Science, Masaryk University, Kotlářská 2, 611 37 Brno, Czech Republic; 3Faculty of Science, J.E. Purkyně University, Pasteurova 15, 400 96 Ústí nad Labem, Czech Republic; 4Institute of Physics of Materials, The Czech Academy of Sciences, Žižkova 22, 616 00 Brno, Czech Republic; 5Department of Physical Electronics, Faculty of Science, Masaryk University, Kotlářská 2, 611 37 Brno, Czech Republic; 6Department of Theoretical and Experimental Electrical Engineering, Faculty of Electrical Engineering and Communication, Brno University of Technology, Technická 12, 616 00 Brno, Czech Republic

**Keywords:** electrospinning, PCL nanofibers, PP fabric, composite, adhesion, low-pressure plasma modification, atmospheric pressure plasma jet, loop test, tensile test

## Abstract

Excellent adhesion of electrospun nanofiber (NF) to textile support is crucial for a broad range of their bioapplications, e.g., wound dressing development. We compared the effect of several low- and atmospheric pressure plasma modifications on the adhesion between two parts of composite—polycaprolactone (PCL) nanofibrous mat (functional part) and polypropylene (PP) spunbond fabric (support). The support fabrics were modified before electrospinning by low-pressure plasma oxygen treatment or amine plasma polymer thin film or treated by atmospheric pressure plasma slit jet (PSJ) in argon or argon/nitrogen. The adhesion was evaluated by tensile test and loop test adapted for thin NF mat measurement and the trends obtained by both tests largely agreed. Although all modifications improved the adhesion significantly (at least twice for PSJ treatments), low-pressure oxygen treatment showed to be the most effective as it strengthened adhesion by a factor of six. The adhesion improvement was ascribed to the synergic effect of high treatment homogeneity with the right ratio of surface functional groups and sufficient wettability. The low-pressure modified fabric also stayed long-term hydrophilic (ten months), even though surfaces usually return to a non-wettable state (hydrophobic recovery). In contrast to XPS, highly surface-sensitive water contact angle measurement proved suitable for monitoring subtle surface changes.

## 1. Introduction

Composites, produced from two or more constituent materials whose combination creates a synergistic effect, have attracted particular interest in recent years because of their unique properties and immense opportunities for modern development and advanced material research. In this paper, we studied a composite material composed of two nonwoven textile layers—a support textile and nanofibers (NFs) applied on it as a top functional layer. Although electrospun NFs can be prepared at a high cost as a self-supporting material, deposition on a supporting textile reduces expenses and benefits from good structural and mechanical support of nonwoven textiles without altering the desirable features of NFs.

NFs are characterized by a high surface area–to–volume ratio and controlled porosity. They also resemble the extracellular matrix surrounding cells by their fibrillar structure. Due to their versatility, NFs have evinced great attention as a potential biomaterial in a broad range of biomedicine applications, e.g., tissue engineering [1,2,3,4,5,6,7], biosensors [8,9,10,11], wound dressings [12,13,14] or drug delivery [15]. Electrospinning is one of the most widely used techniques for generating micro/nanofibers because it is a simple, cost-effective, flexible, and industrially easily up-scalable method. The electrospun NFs are formed in a controlled manner with desired diameters and uniform or special microscopic morphology from a wide range of polymers [16,17,18,19,20]. Synthetic biodegradable polymers, such as polycaprolactone (PCL), poly(l-lactic acid), poly(glycolic acid), and poly(lactic-co-glycolic acid), offer easier processability for electrospinning and more controllable nanofibrous morphology than natural polymers [21]. In this study, PCL was selected because it is a biodegradable polymer used in many FDA-approved surgical implants and drug-delivery devices for tissue engineering and regenerative medicine [22]. It possesses good mechanical properties and long-term stability in vivo [23].

For bioapplications where using a nanofibrous mat together with nonwoven textile support is necessary, i.e., as a wound dressing, good adhesion between individual composite parts becomes crucial. However, spunbond polypropylene (PP) textile, often used as the support and collector during the electrospinning of NFs, is hydrophobic, which can prevent nanofibers from adhering. Adhesion of NFs to textile support can be improved chemically [24] or thermally [25], but plasma processing of material surfaces has started to attract attention in recent years. Plasma technology for surface modifications has many advantages compared to wet chemical treatments, such as low toxicity, short one-step easily tunable fabrication, substrate independence, and, if required, a negligible degradation of the original material. It has been demonstrated in numerous studies [26,27,28,29] that plasma treatment of polymer materials in discharges of gases containing oxygen or nitrogen can introduce polar functional groups on the surface. As a result, the surface free energy of polymer material increases, and a hydrophilic surface is created. Moreover, the polar functional groups can increase material biocompatibility or can be used to graft additional functionalities. However, due to polymer restructuring, this functionalization of a near-surface thin layer has a short duration [30]. In contrast, a coating of material with plasma polymers, i.e., by plasma-enhanced chemical vapor deposition of thin organic films, can produce a large amount of stable functional groups [31,32].

The plasma modification (treatment or deposition) of spunbond textile surface can be efficiently carried out by both the low [33] and atmospheric [34,35,36,37,38] pressure plasma discharges. The advantage of low-pressure plasma modification lies in the process controllability—the surface is modified homogeneously by the desired type of functional groups. However, this approach is ill-suited for industrial applications because of the expensive vacuum systems that are difficult to incorporate into process lines. Therefore, the cheaper non-thermal atmospheric pressure plasma discharges capable of working in the ambient air atmosphere naturally became a more promising option, even though the surface modification is less controllable and less uniform. For textile modification, dielectric barrier discharges (DBDs) are most commonly used [34,35,36,37,38].

Atmospheric pressure plasma jets (APPJs) are a possible alternative to DBDs. The advantage of APPJs over DBDs is the adjustable sample placement distance, which is a simple way of controlling plasma processing conditions. The majority of commercial APPJs (e.g., from Plasmatreat, AFS, SurfaceTreat or Ahlbrand) are based on the transitional arc discharge, a thermalized arc that is cooled down by a gas flow into a non-thermal cold plasma. Although the plasma gas temperature of the non-thermal part of the discharge is low enough for treating the bulk PP samples [39], it is still high enough to melt its surface locally, making the transitional APPJs unsuitable for the modification of thin, thermally sensitive materials such as nonwoven textiles. For such cases, low-temperature APPJs generating the plasma that never reaches the local thermal equilibrium need to be used.

The excellent adhesion between the functional nanofibrous and supporting composite parts is a key property, especially for wound dressing development. On that account, our work aimed to enhance the adhesion of PCL NFs to spunbond PP fabric modified by the radio-frequency (RF) low-pressure capacitively coupled plasma discharge and the low-temperature atmospheric-pressure RF plasma slit jet (PSJ) [40,41]. In a few studies [24,25,33,34,35,38,42], adhesion between NFs and textile support is evaluated, and all of them have employed tensile T-peel tests requiring a rather thick nanofibrous mat. Nevertheless, interest in fabricating thin nanofibrous mats that still retain functionality will grow due to the necessity of cost reduction. To measure the adhesion of a thin nanofibrous mat, we adapted the loop test initially designated for the adhesion measurement of adhesive tapes [43,44,45], which has huge potential as an alternative to the tensile test currently lacking on the market. Therefore, we compared the tensile and loop adhesion test results and discussed their applicability.

## 2. Materials and Methods

### 2.1. Plasma Modifications of PP Spunbond Fabric

The light blue polypropylene (PP) nonwoven spunbond fabric(thickness of (121±3) µm, density of (0.25±0.02) g/cm${}^{3}$, and area density of (30±3) g/m${}^{2}$) was purchased from Pegatex^®^ S (PFNonwovens a.s., Krnov, Czech Republic). The PP fabric was modified in two types of radiofrequency (RF) plasma discharges, low pressure capacitively coupled plasma (CCP) and atmospheric pressure plasma slit jet (PSJ). At low pressure, the PP fabric was modified by oxygen plasma treatment or by plasma-enhanced chemical vapor deposition (PECVD) of amine plasma polymer (amine-pp) film from cyclopropylamine (CPA, 98%, Merck, Darmstadt, Germany) vapors mixed with argon. The modifications were carried out in a ultra-high vacuum stainless-steel reactor with parallel plate electrodes 210 mm in diameter, described in detail in previous publications [46,47]. The bottom electrode was capacitively coupled to a RF generator working at a frequency of 13.56 MHz. The gases were fed into the chamber through a grounded upper showerhead electrode. The distance between the electrodes was 55 mm. The reactor was pumped down to $10^{-5}$ Pa by a rotary vane and turbomolecular pump. The leak rate, including wall desorption, was below 0.01 sccm for all modifications. The pressure was kept constant at 50 Pa throughout both modifications.

The PP fabric was placed on the bottom RF electrode and sputter-cleaned in pulsed plasma with Ar flow of 10 sccm for 5 min. The pulse setting was 33% duty cycle (D.C.) and 500 Hz pulse repetition frequency at 50 W. For oxygen treatment, the PP fabric was exposed to oxygen discharge (continuous wave, 35 W) for 10 min. The O${}_{2}$ flow rate was set to 10 sccm by an electronic MKS flow controller. Amine pps were deposited from a CPA/Ar mixture in a pulsed wave with the same setting as in the case of clean sputtering at 35 W. The Ar flow rate was set to 10 sccm and was regulated by an electronic MKS flow controller, while the CPA vapor flow rate was set to 1 sccm by a needle valve. The deposition time was 30 min.

At atmospheric pressure, the PP fabric was treated with RF PSJ [40,41] with a working frequency of 13.56 MHz at a power of 500 W in ambient air. Two gas feeds were used: Ar and Ar+N${}_{2}$. The Ar flow rate was set to 67 slm and N${}_{2}$ flow rate to 1.5 slm. The distance between the slit exit and the substrate was 10 mm. The PP fabric was placed on a conveyor belt moving at the speed of 100 mm/s. One pass was used.

### 2.2. Electrospinning of PCL Nanofibers

Polycaprolactone (PCL) nanofibrous mats were prepared by electrospinning from the PCL solution on a reference untreated and plasma-modified PP fabric that was placed on the electrospinning collector—a rotating cylinder with a rotation speed of 20 rpm. The PP fabric was used as received. PCL pellets (Mn 80 000, Merck, Darmstadt, Germany) were dissolved in a mixture of acetic acid (99%, Merck, Darmstadt, Germany) and formic acid (98%, Merck, Darmstadt, Germany) in a weight ratio of 2:1 to acquire a PCL solution of 14 wt.% concentration. After mixing, the solution was stirred at room temperature for 24 h. The electrospinning process was performed on INOSPIN Mini device (Inocure, Praha, Czech Republic) based on needle spinning. The electrospinning conditions have been optimized by Kupka et al. [48] and adapted to INOSPIN Mini device. The applied voltage was +40 kV on the needle and −10 kV on the collecting electrode. The electrode distance was 155 mm. The needle used has a diameter of 16 G, and the flow of the polymer solution was 0.35 mL/min. The volume of polymer solution used for one PCL mat was 3 mL. The electrospinning followed within one week after the plasma modification of the PP fabric.

### 2.3. Surface Characterization

The surface morphology of untreated and modified PP fabrics and PCL nanofibers (NFs) was studied by scanning electron microscopy (SEM) using an SEM LYRA3 (Tescan, Brno, Czech Republic) microscope in secondary emission mode (10 kV acceleration voltage, 9 mm working distance). The micrographs were acquired with a resolution of 1024×1024 pixels. Prior to imaging, the samples were coated with a 10 nm thick gold film deposited by RF magnetron sputtering (Leica ACE 600, Leica Microsystems, Wetzlar, Germany) to eliminate the charging of the sample surface during imaging.

X-ray photoelectron spectroscopy (XPS) used for the surface chemical characterization (information depth from the maximum of 5 nm) was carried out using an Axis Supra (Kratos Analytical, Manchester, UK) spectrometer with X-ray monochromated source (combined Al/Ag anode). To avoid differential charging of samples, spectra were acquired with a charge neutralization in overcompensated mode. The binding energies were corrected by shifting the hydrocarbon component CH${}_{x}$ to 285.0 eV. The elemental atomic percentage was quantified from the high-resolution spectra of each element taken at the pass energy of 20 eV. The high-resolution C 1s spectra were fitted by CasaXPS software version 2.3.19 (Casa Software Ltd., Teignmouth, UK) after subtracting the Shirley-type background to obtain individual components. The C–C/CH${}_{x}$ had an asymmetric peak shape that was derived from the spectrum of pure PP [39,40]. The other components were Gaussian–Lorentzian (G-L) shaped with a fixed G-L percentage of 30%. The full width at a half maximum was between 0.9–1.3 eV for all components. The values of the binding energies for various carbon environments were taken from the literature [49,50,51,52].

The water contact angle (WCA) values were measured by the sessile drop (3 µL of demineralized water) method and evaluated by SeeSystem 7.0. (Advex Instruments, Brno, Czech Republic) software calculating the WCA on the basis of three-point interpolation of the drop height and width. The mean WCA and its standard deviation were estimated from ten droplet measurements.

### 2.4. Adhesion Testing of PCL NFs to PP Fabric

For the tensile T-peel adhesion tests (called also peel or peeling tests), adopted from ASTM D1876, 10 mm wide and 60 mm long strips were cut from the NF-coated PP fabric with a scalpel. The nanofibrous mat (white) and the fabric (light blue) were carefully separated from each other in a length of about 10 mm and glued to the aluminium holders by thin double-sided tape (Figure 1a). The holders were then attached using pins to the jaws of the testing machine MTS Tytron 250 Microforce system (MTS Systems Corporation, Eden Prairie, MN, USA), equipped with a dynamometer designed for accurate measurement of small forces (max. 10 N). The jaws of the testing machine were moved apart (Figure 1b). The tests were performed in a constant piston speed mode (0.33 mm/s). The time, piston position, and force were recorded. The rupture events between the PCL mat adhering to the PP spunbond fabric are represented by multiple peaks visible in Figure 2, where an example of a typical tensile T-peel adhesion test curves is shown. At least four samples were tested for each type of material. The maximum force values for individual rupture events were determined. Only peaks satisfying the following criteria were included: (a) maximum force exceeding three times the noise value and (b) close peaks could be distinguished with confidence. Obtained data were averaged and expressed as the mean peak force and its standard deviation.

The loop adhesion tests were performed by measuring the force required to tear the electrospun nanofibrous mat from the defined area of PP fabric. A TA.XTplusC Texture Analyzer (Stable Micro Systems, Godalming, UK) with a 5 N load cell was used for the measurement. The loop test is based on the ASTM D6195 standard for the adhesion measurement of adhesive tapes. Adhesive tape Elcometer (Elcometer, Manchester, UK) ISO 2409:2003 was used in all the measurements. The substrate side of the sample cut into 25×25 mm pieces was glued with double-sided adhesive tape to a plastic holder, which was then fixed in the lower jaw of the measuring device. A test adhesive tape was placed in a controlled manner in the upper movable jaw to form a loop (Figure 1c). Keeping the same length of 100 mm and the shape of the loop, a reproducible pressure force of the test adhesive tape on the sample was achieved (Figure 1d). Subsequently, the jaws were moved away, and the magnitude of the applied force versus jaw position was recorded (Appendix A). The movement speed of the jaw head was 5 mm/s, and the trigger force was 10 mN. Each sample was measured at least ten times.

To allow direct comparison between various treatments and testing methods Work per Area (W/A) was calculated as work done by the peeling force divided by the peeled area. It can be calculated for both tests, unlike the mean peak force which has no loop test equivalent. Mean peak force is also more strongly surface dependent and sensitive to local imperfections because they are evaluated from individual rupture events.

## 3. Results and Discussion

### 3.1. Characterization of As-Modified PP Fabric

The PP fabric used as a supporting part of the composite was a net of randomly oriented fibers, (21.2±0.3) µm in diameter (Figure 3a). It was not composed only of pure polypropylene because, in addition to carbon, XPS analysis revealed the presence of oxygen ((17.3±0.2) at.%), potassium ((4.9±0.1) at.%), and phosphorus ((3.2±0.1) at.%). These elements originated from the fabrication process of spunbond. An initial high amount of oxygen, O/C = 0.233 (Figure 4a), was derived from numerous carbon-bonded oxygen-containing surface groups as seen in the fitting of C 1s high-resolution spectra. Apart from aliphatic hydrocarbon groups (CH${}_{x}$, C–C) at 285.0 eV, the following groups were identified: C–COO at 285.8 eV, carbon single bonded to oxygen (C–O) at 286.7 eV, carbon double bonded to oxygen (C=O) at 287.5 eV, and carboxyl/ester (C(O)OR) at 288.7 eV (Figure 5e). The structure and chemical composition of PP fabrics after different types of plasma modification were studied to obtain complex information about effects on the PP fabric properties which are crucial for understanding the adhesion of PCL NFs.

After low-pressure modifications, SEM showed that neither oxygen treatment nor amine-pp deposition altered or damaged the fibrous structure of the nonwoven PP fabric. The appearance was identical to the reference PP fabric shown in Figure 3a. The amount of oxygen did not significantly change; the O/C ratio increased by about 0.06 from 0.23 to 0.29 compared to untreated PP fabric (Figure 4a). It corresponds to a rise of atomic oxygen by about 4 at.%. The distribution of oxygen functional groups was different for PP fabric before and after oxygen treatment (compare Figure 5a,e). The amount of all oxygen-containing groups determined based on the fitting of XPS C 1s high-resolution spectra was the same, around 3 at.%, with the exception of carbon single bonded to oxygen (C–O) whose concentration was 1.5× higher. Moreover, the fitting model for C 1s high-resolution spectra had to be modified to correspond to the fabric after oxygen treatment. The positions of C–O, C=O and CO(O)R components were shifted to higher binding energies of 287.0, 288.2 and 289.3 eV, resp. (Figure 5a). A small amount of nitrogen (around 1 at.%) was detected on the surface of the PP fabric after the oxygen plasma treatment. It comes from the cross-contamination related to the amine-pp deposition performed in the same plasma reactor.

The successful coating by the amine-pp film was confirmed by the presence of nitrogen (N/C ratio 0.14, corresponding to 9.3 at.% of nitrogen) on the fabric after the deposition (Figure 4a) because nitrogen is not part of the original fabric. Moreover, there is a good agreement between the shape and fitting of C 1s peak and the already published amine pp results [50,53]. The C 1s high-resolution peak of amine pp was composed of five components corresponding to the following groups: aliphatic hydrocarbon groups (CH${}_{x}$, C–C) at 285.0 eV, amino groups bonded to carbon (C–NH${}_{x}$) at 285.9 eV, imine or nitrile groups (C=N/C≡N) at 286.7 eV, aldehyde/ketone or amide groups (C=O/N–C=O/N–C–O) at 287.9 eV, and ester/carboxyl groups (C(O)OR) at 288.9 eV (Figure 5b). The sum of all nitrogen and oxygen-containing functional groups was almost 20 at.% after amine-pp deposition. Generally, the surface of plasma polymers is not yet stable directly after the deposition due to the formation of short-lived reactive species and radicals that react with oxygen after exposure to ambient air [54,55,56]. Thus, oxidation is responsible for the presence of oxygen in amine pp (the O/C ratio = 0.26).

From the industrial point of view, upscaling of low-pressure plasma modifications, e.g., roll-to-roll processing, can be expensive and hardly pay off. Therefore we proceeded with modifications of PP fabric by plasma at atmospheric pressure, which does not suffer from such drawbacks and thus is more perspective from the application point of view. At atmospheric pressure, the PP fabrics were functionalized by RF PSJ fed with Ar gas or a mixture of Ar and N${}_{2}$ in open ambient atmosphere. Contrary to the low-pressure modifications, the atmospheric jet slightly affected the structure of the PP fabric (Figure 3). When we compare untreated PP (Figure 3a) to textiles after treatments, we can notice fiber merging after the Ar treatment (Figure 3b) and local melting in the case of Ar+N${}_{2}$ treatment (Figure 3c). The defects likely arose due to the prolonged PSJ filament contact that induced a higher thermal load on the PP fibers.

Both treatments led to an increase in oxygen concentration on the PP spunbond fabric (Figure 4a) through the oxidation of discharge activated surface in ambient air. While the O/C ratio of the original fabric was 0.23, the O/C ratio after Ar and Ar+N${}_{2}$ treatments reached values higher than 0.34 and 0.39, respectively. In the case of low-pressure oxygen treatment, the O/C ratio increased just to 0.29. Thus atmospheric pressure modifications turned out to be more effective in introducing oxygen into the surface of the PP fabric. Since the reactivity of nitrogen gas species with the PP chain is much lower than of the oxygen reactive gas species [40,57], the modified fabrics contained only below 1 at.% of nitrogen regardless of the PSJ gas feed. The C 1s high-resolution spectra of treated textile was fitted by the same five-component model used for the untreated textile spectra (Figure 5c–e). The content of identified groups directly after both jet treatments was similar. Moreover, ratios among oxygen-containing groups were comparable for both fabrics treated with atmospheric PSJ and untreated PP fabric (approximately C–COOR:C–O:C=O:COOR = 10:5:3:1).

Wettability is another major factor characterizing the surface properties of materials. Despite a non-negligible amount of oxygen-containing polar groups (16 at.% in total (Figure 5e), the PP fabric was highly hydrophobic with WCA of 127.2°±0.8° (Figure 4b and Appendix A). Meaning polar groups were oriented mainly toward the bulk of the material and did not affect the top surface properties. Although atmospheric PSJ treatments led to a significant increase in oxygen concentration, the WCA of PP fabric decreased by only 10–13° and the surface stayed hydrophobic. As mentioned above, the plasma of atmospheric PSJ was filamentary, and thus the treatment was not perfectly homogeneous. Small local domains of PP fabric could remained untreated, affecting the measured WCAs highly sensitive to the lower-energy part of surface areas [58]. A slightly less hydrophobic surface was achieved after treatment with the Ar+N${}_{2}$ mixture (113.7${}^{{}^{\circ}}$ versus 116.5°, Appendix A) which can be associated with a higher amount of oxygen compared to the Ar treatment, but the difference is not significant. The bottom side of the PP fabric stayed unaltered, keeping the same WCA as before treatment, confirming that the PSJ treatment affected just the uppermost surface in contact with the plasma, and discharge filaments did not propagate inside or below the fabric.

In contrast to atmospheric PSJ treatments, both low-pressure plasma modifications resulted in PP fabric with highly hydrophilic properties; the WCA was below the measuring limits of the used method, i.e., lower than 10° (Appendix A). The PP fabric was hydrophilic even from the untreated bottom side, evidencing high penetration depth of low-pressure modifications. It is obvious that there is no direct correlation between the number of oxygen-containing groups and surface wettability. Although the PP fabric treated with low-pressure oxygen plasma contained a lower amount of polar groups when compared to the atmospheric plasma treatments, it was the most hydrophilic. The local homogeneity changes and possibly penetration depth of the treatment, together with the orientation of functional groups towards the material surface or bulk, all likely influencing the measured WCA, were not detectable by the area-averaged XPS. Therefore, surface wettability did not correspond with the polar group content.

### 3.2. Time Stability of PP Fabric Plasma Modifications

The plasma modification of PP fabric precedes the electrospinning of nanofibers during the composite preparation. It can be either part of the same process or carried out in two separate steps (our case). In a two-step process, it is necessary to know how long after the modification, the electrospinning of NFs can still benefit from the activated functionalized surface. Therefore, the time stability of different plasma modifications of PP fabric in terms of the changes in chemical composition and WCA was monitored for more than two weeks (Figure 4b and Figure 6).

As mentioned above, plasma polymers (amine pp sample) oxidize after exposure to the ambient atmosphere. In general, the number of oxygen-containing functional groups increases at the expanse of other functional groups, e.g., nitrogen-containing groups. This effect was not too prominent in the case of amine-pp coated PP fabric (Figure 6b). Ten days after deposition, nitrogen concentration decreased by about 1.5 at.% while oxygen increased by about 3 at.%. After that, the composition of amine pp stopped evolving. Still, both types of groups are hydrophilic and should positively impact the adhesion of PCL nanofibers.

The elemental composition of PP fabric treated with atmospheric PSJ in the mixture of Ar+N${}_{2}$ stayed constant during the monitored period (Figure 6d). On the other hand, the Ar treatment was not completely stable as the oxygen content decreased gradually from 22 to 16 at.% starting ten days after the treatment (Figure 6c). In the case of the PP fabric treated with low-pressure oxygen plasma (Figure 6a), we observed a slight increase in oxygen concentration during the first four days after treatment instead of an expected decrease. Free radicals introduced by oxygen plasma can be responsible for the initial rise of oxygen. Short-lived radicals can activate aliphatic carbon groups, reacting with oxygen in the air. However, they are consumed quickly. After the first four days, no other changes in composition occurred.

The functional groups introduced by plasma treatments typically undergo a process called hydrophobic recovery, through which they reorient themselves towards the bulk of the material, quickly vanishing from the surface [30,59]. However, results from XPS analyses of PP fabric aging (Figure 6a,d) did not agree with this general behavior. We concluded that the hydrophobic recovery is still happening, but such subtle change is difficult to observe by XPS which has information depth of approx. 5 nm and the aging should at first occur close to the surface where the chemical gradient is the steepest. The conclusion can be supported by findings drawn from the wettability study.

In contrast to XPS, WCA is a strongly surface-based method highly sensitive to changes in surface functional groups. Therefore, subtle changes can be easier detected by WCA measurement than by XPS. The PP fabric modified by low-pressure oxygen treatment and amine-pp deposition stayed highly hydrophilic, even after ten months, which is in good agreement with the fast stabilization of composition shown by XPS. A long-time stable hydrophilic modification was not expected, particularly after oxygen treatment. Nevertheless, it can be explained by the deep penetration of low-pressure modifications into the fibrous PP fabric. As a result, there is no significant gradient in functional group concentrations that could give rise to the further migration of surface functional groups towards the bulk of the material.

A more interesting development was noted for atmospheric PSJ treatments. WCAs of modified PP fabric (Figure 4b and Appendix A) started to rise immediately after both jet treatments but did not return to the original value during the monitored period of 15 days. The aging was more rapid for Ar+N${}_{2}$ treatment which also showed larger WCA variances. The variable WCA can be associated with higher inhomogeneity of treatment after adding nitrogen to PSJ compared to Ar gas feed [40]. Moreover, the structure of PP fabric suffered more severe local defects after Ar+N${}_{2}$ treatment, as apparent in Figure 3c, which can also cause WCA variability. Despite different initial WCAs of jet-treated PP fabric, the wettability reached the same values within the margin of error (≈119.5°) eight days after the treatments. The changes in surface composition were so prominent that they started to be detectable even by XPS in the case of Ar PSJ 10 days after treatment (Figure 6c). We assume PSJ modified only the upper-most layer of the surface, creating a gradient in surface functional groups, which drove fast migration of functional groups to the bulk of the material. The different penetration depths of low and atmospheric pressure modifications thus explain different aging rates evident from WCA changes.

### 3.3. Adhesion of PCL NFs to PP Fabric

The PP fabrics modified by low-pressure plasma (oxygen treatment and amine-pp coating) and atmospheric pressure PSJ (Ar and Ar+N${}_{2}$ treatment) were used as the support for the electrospinning of PCL NFs. The thickness of the resulting nanofibrous mat was about 60–80 µm depending on the electrospinning batch. The rather high thickness and sufficient stiffness were suitable for tensile tests preventing the tearing of the nanofibrous mat during testing. The nanofibrous surface of the composites was analyzed by SEM (Figure 7). The mats were compact, without defects with homogeneous NFs ((218±5) nm in diameter).

First, the PCL NFs electrospun on untreated and plasma-modified PP fabrics were subjected to the tensile T-peel adhesion test. The mean peak force (see Section 2.4 for definition) needed to rip off the PCL mat from the PP fabric is shown in Figure 8, the reference value was (0.025±0.002) N. We confirmed that the force needed to separate the PCL mat increased with the introduction of functional groups onto the PP fabric surface and was higher for all plasma-modified PP fabrics. In the case of low-pressure modifications, the amine pp film increased the mean peak force by only about 0.01 N, but O${}_{2}$ treatment increased it more than 3×. Both atmospheric pressure modifications (Ar and Ar+N${}_{2}$) improved the mean peak force almost 2×.

The aging of treatments represented by changes in functional group concentrations can influence adhesion. Since the Ar+N${}_{2}$ treatment aged the fastest, it was selected for the time-dependent adhesion test. Electrospinning batches prepared one and eight days after Ar+N${}_{2}$ PSJ treatment were compared, and the results are shown in Figure 8 (1D and 8D). The mean peak forces were the same within the error. Although the WCA values for atmospheric pressure modifications tend to return to the original values, the increase of 6° for Ar+N${}_{2}$ treatment over the course of one week was not sufficient to change the adhesion.

The results for work per area (W/A) are shown in Figure 9—left. The reference value for tensile tests was (2.3±0.4) N/m. In the case of low-pressure O${}_{2}$ treatment, W/A was almost 5× higher than for the reference. The overall trends in adhesion expressed by mean peak force and W/A were similar for all plasma modifications. However, there were differences since W/A characterizes the adhesion of the entire surface and measures all connections between the NF mat and the PP fabric, whereas the mean peak force measures only a subset of the high-quality connections. It was confirmed that there is no significant difference in adhesion when the electrospinning is performed 1 or 8 days after plasma modification by Ar+N${}_{2}$ treatment. Based on this finding, the adhesion can be considered unaffected by aging if electrospinning is performed within a week after modification.

The tensile test requires a rather thick nanofibrous mat to avoid damage, whereas the general effort for cost reductions can profit from electrospinning of as thin a nanofibrous mat as possible that still retains functionality. To measure a thin nanofibrous mat, an adapted loop adhesion test can be employed. In the loop test, the adhesion of PCL mats to the PP fabrics was again expressed as W/A (Figure 9—right). In this case, the nanofibrous mats were electrospun in two batches. Hence, two different references were used, one for each batch. Reference 1 ((1.69±0.15) N/m) corresponds to composites modified in low-pressure plasma and reference 2 ((0.61±0.05) N/m) corresponds to atmospheric pressure treated PP fabric. Averaging all reference loop test results together would give W/A of (1.03±0.12) N/m (not shown in Figure 9—results for composites must be compared to the corresponding references). As the difference in reference values illustrates, the electrospinning process itself can influence adhesion because it determines the quality of the resulting fibers, i.e., their diameter, homogeneity, porosity, etc. The impact of electrospinning cannot be neglected, as it directly affects the fibers’ ability to form interconnections with the modified PP fabric and the number of such bonds.

The most effective modification according to the loop test was again the low-pressure oxygen treatment which improved the adhesion 6×. The absolute W/A values from both tests were similar, (11.3±1.3) N/m (tensile) and (12.1±1.5) N/m (loop) although it was partly a coincidence since the reference differed. In the case of atmospheric PSJ treatments, the absolute W/A values were significantly lower compared to tensile test results. Still, the improvement was 2.5× for Ar treatment and nearly 2× for Ar+N${}_{2}$ treatment, thus following the same trends as tensile tests. Thus trends obtained by both mechanical testing methods largely agreed. We can conclude that either is suitable for characterization of adhesion between NFs and a fabric support. Moreover, the loop test can be the only option for thin nanofibrous mats whose adhesion determination has so far been difficult.

The differences in adhesion results for plasma modifications used cannot be satisfactorily explained by XPS analyses alone. From the chemical point of view, the hypothesis can be that presence and number of oxygen-containing and nitrogen-containing functional groups after plasma modifications is the key to improving the PCL mat adhesion to the PP fabric. The total amount of neither carbon nor CH${}_{x}$ groups can be important parameter because they are almost the same for reference and O${}_{2}$ treatment despite huge improvement in adhesion. The only notable difference was in the number of ester/carboxyl groups, which increased 3.5× after O${}_{2}$ treatment. Low-pressure O${}_{2}$ treatment also led to considerably better results than atmospheric PSJ treatments, even though the O/C ratio was around 0.28 for O${}_{2}$ treatment and between 0.31 and 0.34 for Ar and Ar+N${}_{2}$ modifications. In the case of amine pp, the O/C and N/C ratios were 0.26 and 0.14, respectively. The total amount of functional groups was high, but the adhesion between the composite layers was not so pronouncedly improved. Thus, we concluded that the presence of oxygen-containing functional groups is more beneficial for improving the PCL mat adhesion to the PP fabric than the presence of nitrogen-containing groups in amine pps.

The interconnection between the PP fabric and PCL NFs occurs directly on the interface between the composite layers. Therefore, the WCA development corresponding to changes in free surface energy is more relevant than 5 nm depth-averaged chemical composition given by XPS. As mentioned before, O${}_{2}$ and amine-pp modified PP fabrics were completely wettable (i.e., with high free surface energy) with WCA lower than 10° (Figure 4b). Therefore, a synergistic effect on adhesion occurred when sufficient wettability was combined with a suitable ratio of functional groups on the surface. As mentioned above, the XPS-measured ratios of oxygen-containing groups for both fabrics treated with atmospheric PSJ and the untreated PP fabric were comparable. Therefore, the grafted oxygen-containing groups responsible for the adhesion increase were observable only indirectly by the wettability measurements. Furthermore, the electrospinning process of a dense fibrous network at the nanometer scale is very sensitive to the surface domains and imperfections of the collector—PP fabric. Consequently, a lower amount of high-quality interconnections can form in the case of atmospheric PSJ modifications because of worse homogeneity and local defects caused by treatment. The high surface homogeneity with the right ratio of surface functional groups and sufficient wettability is crucial for adhesion improvement between the PP fabric and PCL NF mat. The low-pressure O${}_{2}$ treatment satisfied these conditions best.

A limited number of studies have dealt with plasma enhancement of electrospun NF adhesion to the supporting textile; most were conducted in atmospheric dielectric barrier discharge (DBD). The adhesion of polyamide NFs or hybrid Ag/polyacrylonitrile NFs to supporting textile increased 2–3× after DBD in air [34,38] or He and He/O${}_{2}$ mixture [35]. Treatment of cotton fabric in He and He/O${}_{2}$ DBD before electrospinning of chitosan NFs enhanced average adhesion strength almost 4×. The better outcomes after low-pressure oxygen treatment agree with the work of Romabaldini et al. [33], where adhesion energies and forces were about 5–15× higher for polyethylene oxide nanofibrous mats when deposited on plasma-treated PP fabric but only about 2–3× higher for polyamide NFs. From widely varying results among different research groups, it is evident that the right combination of NF material, support fabric structure, and type of plasma treatment must be found in each application. Further, more detailed comparative research needs to be conducted to fully understand all the factors governing the adhesion of electrospun NFs to the textile support.

The results reported in DBD are comparable with the improvement of adhesion after atmospheric PSJ in our case. However, in contrast to DBD, the properties of PSJ discharge can be controlled by simple adjustment of sample placement distance. Moreover, compared to stationary filaments of DBDs, the PSJ filaments changed positions over time, shuffling along the length of the slit at the jitter speeds in orders of m/s [41]. This constant motion ensures higher treatment homogeneities even for higher sample movement speeds, i.e., low treatment times. Nevertheless, the surface properties are still much more homogeneous after low-pressure modifications.

Although the low-pressure O${}_{2}$ treatment has shown to be the most effective method of PCL NFs adhesion improvement, integration of low-pressure processes in existing productions is challenging and costly. It can suit advanced applications requiring high performance and control over composite properties. However, atmospheric plasma devices are more promising from the industrial point of view as they offer better scalability, low initial investment costs, and can be easily implemented into the production line. Twofold adhesion improvement after atmospheric PSJ treatments can be sufficient for many applications or for making NF handling easier.

## 4. Conclusions

Adhesion between supporting (nonwoven spunbond PP fabric) and functional (PCL nanofibrous mat) parts of the composite was improved by modifying the PP fabric either in low-pressure discharge or atmospheric pressure PSJ before PCL electrospinning. Both plasma modifications enabled the creation of new nitrogen- or oxygen-containing polar groups. The PP fabric was less hydrophobic (WCA decreased by 10°) in the case of atmospheric PSJ treatment but became highly hydrophilic after low-pressure modifications. In contrast to XPS, sensitive surface-based WCA demonstrated a slow return to original values after atmospheric PSJ. The PP fabric after low-pressure modifications stayed highly hydrophilic (WCA $<10^{{}^{\circ}}$) for more than ten months. Different aging rates can be attributed to the penetration depths of low and atmospheric pressure discharges. While atmospheric PSJ modifies only the upper-most layer of the surface followed by functional group migration to the bulk, low-pressure modifications can propagate deep into the PP fabric, so no significant group migration emerges.

All the plasma modifications led to increased adhesion. The trends obtained by tensile and loop tests largely agreed, and either is suitable for the characterization. Moreover, the loop test is a novel option for thin nanofibrous mats whose adhesion determination has been difficult. Low-pressure oxygen treatment was the most effective; the work per area increased by a factor of almost six. Nevertheless, incorporating vacuum devices into an industrial process is expensive and feasible only for special applications, e. g., wound dressings. For less demanding applications, the two times improvement achieved by atmospheric PSJ is sufficient. The adhesion improvement was ascribed to the synergic effect of high surface homogeneity of treatment with the proper ratio of surface functional groups and sufficient wettability.

## Figures and Tables

**Figure 1 polymers-15-01686-f001:**
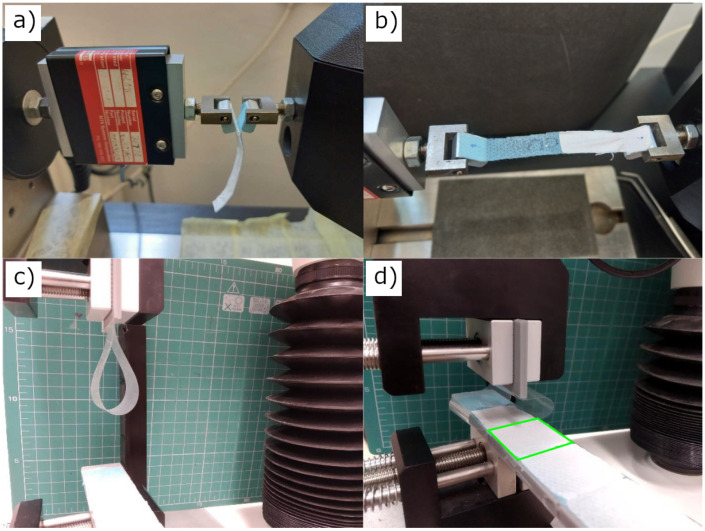
Beginning (**a**) and ongoing (**b**) measurement of the adhesion force between the PCL mat (white) and the PP fabric (blue) by the tensile T-peel test. Beginning (**c**) and ongoing (**d**) measurement of the adhesive force between the PCL mats and the PP fabric by the loop test. The area of one sample is marked with a green square.

**Figure 2 polymers-15-01686-f002:**
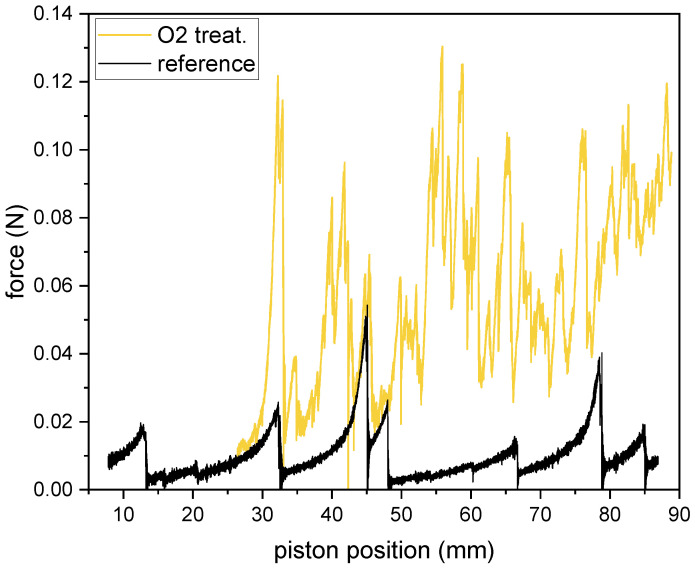
Example of a typical tensile T-peel adhesion test result. The curve shown was obtained for the reference sample and O${}_{2}$ treated sample.

**Figure 3 polymers-15-01686-f003:**
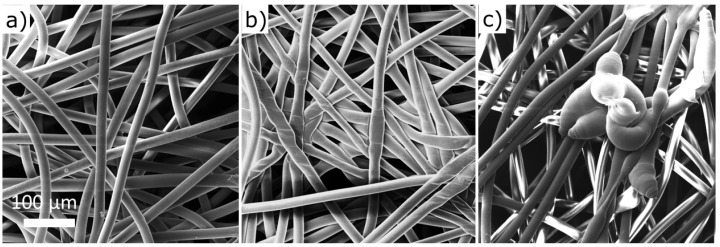
SEM micrographs of PP fabric before treatment (**a**), after Ar treatment (**b**) and after Ar+N${}_{2}$ treatment (**c**); view field of 500 µm.

**Figure 4 polymers-15-01686-f004:**
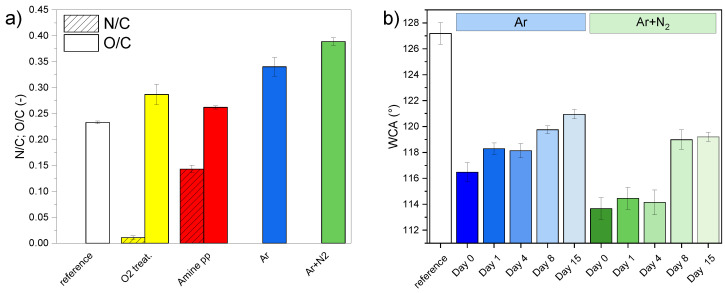
(**a**) XPS chemical composition of PP fabric: as-received (reference), modified by low-pressure plasma (O${}_{2}$ treatment and amine-pp thin film) and atmospheric pressure plasma (Ar and Ar+N${}_{2}$ treatment). (**b**) Changes of water contact angle (WCA) of PP fabric in time: as-received (reference), modified by atmospheric pressure plasma (Ar and Ar+N${}_{2}$ treatment). The WCA after low-pressure plasma modifications was not measurable, i.e., was lower than 10°.

**Figure 5 polymers-15-01686-f005:**
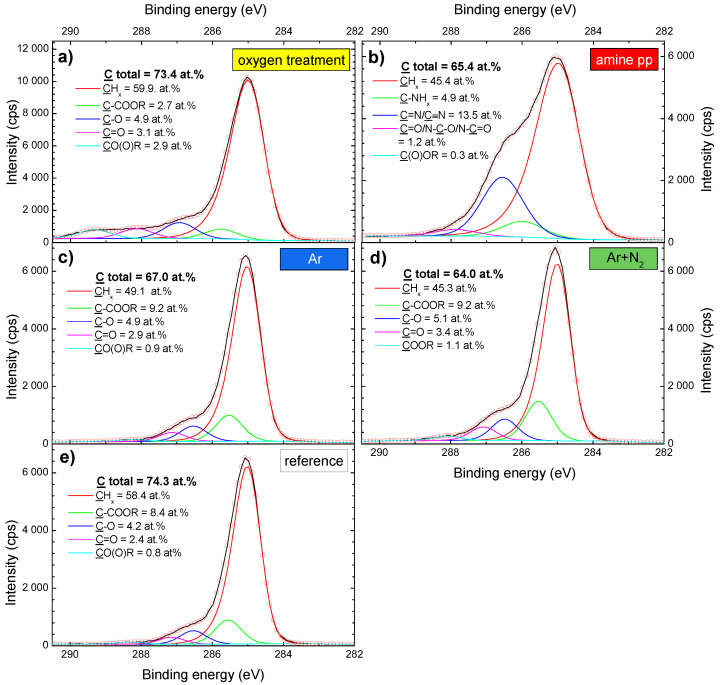
Fitted XPS C 1s high-resolution spectra of PP fabric: modified by low-pressure plasma (oxygen treatment (**a**) and amine pp thin film (**b**)) and atmospheric pressure plasma (Ar treatment (**c**) and Ar+N${}_{2}$ treatment (**d**)), and as-received (reference (**e**)). Stated atomic percentages of chemical groups correspond to PP fabric directly after modification.

**Figure 6 polymers-15-01686-f006:**
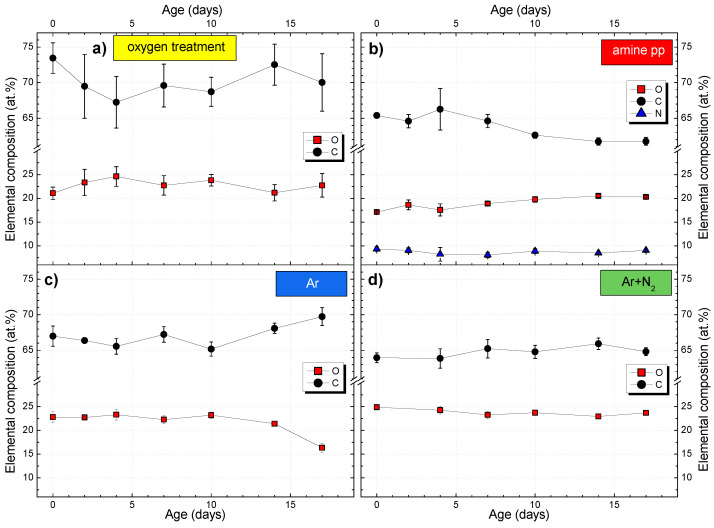
Aging of PP fabric modified by low-pressure plasma (oxygen treatment (**a**) and amine pp thin film (**b**)) and atmospheric pressure plasma (Ar treatment (**c**) and Ar+N${}_{2}$ treatment (**d**)) demonstrated by chemical composition.

**Figure 7 polymers-15-01686-f007:**
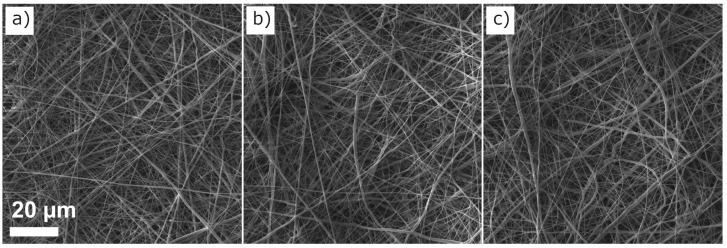
SEM micrographs of PCL NFs elecspun on PP fabric: untreated (**a**), treated in Ar atmospheric RF jet (**b**), treated in Ar+N${}_{2}$ atmospheric RF jet (**c**); view field of 100 µm.

**Figure 8 polymers-15-01686-f008:**
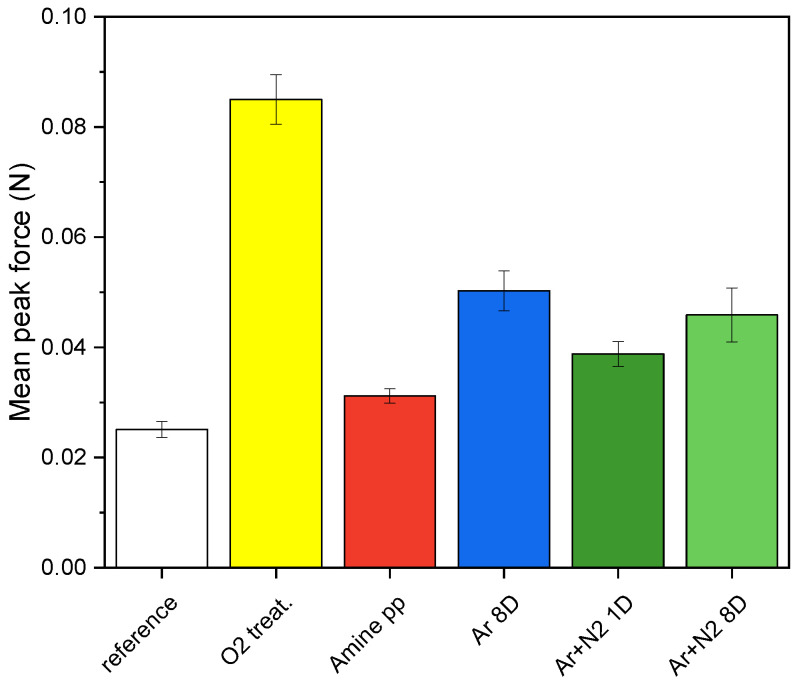
Mean peak force determined by tensile T-peel tests of the PCL mat adhesion to the PP fabric without treatment (reference), modified by low-pressure plasma (O${}_{2}$ treatment and amine pp) and by atmospheric pressure plasma (Ar treated and Ar+N${}_{2}$ treated).

**Figure 9 polymers-15-01686-f009:**
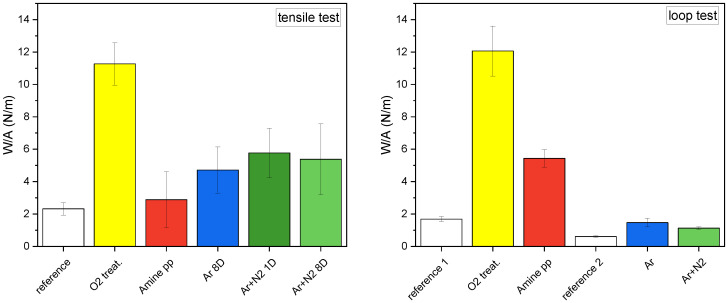
Work per area (W/A) necessary to tear off the PCL nanofibrous mat determined by tensile T-peel tests—**left** and by loop tests—**right**. Reference without treatment, modified by low-pressure plasma (O${}_{2}$ treatment and amine pp) and by atmospheric pressure plasma (Ar treated and Ar+N${}_{2}$ treated).

## Data Availability

The data presented in this study are available in the main text.

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
