# Peer review of "Enhanced Adhesion of Electrospun Polycaprolactone Nanofibers to Plasma-Modified Polypropylene Fabric"

_polymers, 2023, doi:10.3390/polym15071686_

Round 1
Reviewer 1 Report
In this manuscript, authors improved adhesion between electrospun PCL layer and substrate PP fabric, which is pretreated with plasma. Overall, the experiment is well designed and described. Some comments are as follows:
1. Line 30, some medical applications are mentioned regarding NFs, it’s suggested that authors may cite the following paper "A bioinspired, durable, and nondisposable transparent graphene skin electrode for electrophysiological signal detection." ACS Materials Letters 2.8 (2020): 999-1007, in which electrospun fibers are used to transfer graphene to fabric electrodes for sEMG sensing.
2. For ASTM D6195, it’s a standard to determine the peel strength of pressure-sensitive adhesive. However, in the loop test, it’s confusing that how authors can determine when the electrospun fibers were torn from the mat? Please clarify.
3. It’s suggested that author may conduct the contact angle test to visually reflect the wettability.
4. Line 357, authors mentioned thickness of the electrospun mat. Please clarify the elctrospinning duration time as well as the receiver type (it’s a rotating receiver or a plate?). Also, different voltage will result in different fiber diameter, leading to different mat thickness (assuming the duration time is the same), have authors studied the influence of PCL thickness on adhesion?
Reviewer 2 Report
1. The captions under the pictures are very detailed. Part of the information from the captions under the figures should be transferred to section 2 of the article.
2. Figure 1 consists of four blocks, which should be divided into a) b) c) d) and signed separately.
3. It is desirable to split the image in Figure 7.
4. Conclusions should be shortened and specified.
5. The purpose of the work should be formulated clearly for further understanding where the obtained results can be used.
6. Part of the information from the Introduction should be moved to Section 2 of the article (Lines 80-92).
Reviewer 3 Report
This manuscript investigated the low and atmospheric pressure plasma modifications on the adhesion between PCL nanofibrous mat and PP spunbond fabric. Some major revisions should be conducted before publication.
1. Title is a little bit misleading, and it should be rewritten.
2. The Abstract section should be rewritten in a better and clear way. For instance, some important result data like adhesion performance are suggested to be presented in a much more detailed manner.
3. Please state the reasons why PCL and PP were chosen in this study. What are the merits and advantages of them compared with some other biopolymers that are commonly used in wound dressings?
4. The merits of electrospinning technique should be further outlined, and some recent works about the innovative electrospinning like 10.1016/j.eurpolymj.2023.111863 and 10.1021/acsami.1c24131 are suggested to be discussed.
5. The technique parameters of PP nonwovens should be given, like thickness and density.
6. Please state the reasons why 14% (w/w) PCL was utilized for electrospinning. Moreover, how did the authors choose the parameters of electrospinning? Do they conduct any preliminary experiments?
7. The testing parameters of water contact angle test should be provided. It also applies to the other techniques.
8. The actual photographs of water contact angle in each group are suggested to added in Figure 3, in order to increase the readability.
9. How about the mechanical properties of PP nonwovens after the plasma modification? Did the process affect the mechanical properties?
10. The Conclusion section is too long, which should be shortened.
11. The grammar and writing should be improved in the whole manuscript.
Round 2
Reviewer 1 Report
Thanks for the updated manuscript. Concerns have been properly addressed. It can be accepted in present form.
Author Response
We kindly thank the reviewer for approving our manuscript for publication.
Reviewer 3 Report
The reviewer's concerns have been well addressed.
Author Response
We kindly thank the reviewer for the response.